# Role of Altered Metabolism of Triglyceride-Rich Lipoprotein Particles in the Development of Vascular Dysfunction in Systemic Lupus Erythematosus

**DOI:** 10.3390/biom13030401

**Published:** 2023-02-21

**Authors:** Ágnes Diószegi, Hajnalka Lőrincz, Eszter Kaáli, Pál Soltész, Bianka Perge, Éva Varga, Mariann Harangi, Tünde Tarr

**Affiliations:** 1Division of Metabolic Diseases, Department of Internal Medicine, Faculty of Medicine, University of Debrecen, 4032 Debrecen, Hungary; 2Department of Medicine, Västerviks Sjukhus Hospital, 593 33 Västerviks, Sweden; 3Department of Internal Medicine, Faculty of Medicine, University of Debrecen, 4032 Debrecen, Hungary; 4Division of Clinical Immunology, Department of Internal Medicine, Faculty of Medicine, University of Debrecen, 4032 Debrecen, Hungary; 5Department of Internal Medicine and Hematology, Semmelweis University, 1083 Budapest, Hungary; 6ELKH-UD Vascular Pathophysiology Research Group 11003, University of Debrecen, 4032 Debrecen, Hungary

**Keywords:** systemic lupus erythematosus, carotid intima-media thickness, augmentation index, flow-mediated dilation, triglyceride, very low-density lipoprotein, intermediate-density lipoprotein, triglyceride-rich lipoproteins

## Abstract

Background: Impaired lipid metabolism contributes to accelerated inflammatory responses in addition to promoting the formation of atherosclerosis in systemic lupus erythematosus (SLE). We aimed to evaluate the lipid profile, inflammatory markers, and vascular diagnostic tests in active SLE patients to clarify the association between dyslipidemia and early vascular damage. Patients and Methods: 51 clinically active SLE patients and 41 age- and gender-matched control subjects were enrolled in the study. Lipoprotein subfractions were detected by Lipoprint. Brachial artery flow-mediated dilation and common carotid intima-media thickness were detected by ultrasonography. Arterial stiffness indicated by augmentation index (Aix) and pulse wave velocity was measured by arteriography. Results: We found significantly higher Aix, higher VLDL ratio, plasma triglyceride, ApoB100, and small HDL, as well as lower HDL-C, large HDL, and ApoA1 in patients with SLE. There was a significant positive correlation of Aix with triglyceride, VLDL, IDL-C, IDL-B, and LDL1. A backward stepwise multiple regression analysis showed IDL-C subfraction to be the best predictor of Aix. Conclusions: Our results indicate that in young patients with SLE, triglyceride-rich lipoproteins influence vascular function detected by Aix. These parameters may be assessed and integrated into the management plan for screening cardiovascular risk in patients with SLE.

## 1. Introduction

Systemic lupus erythematosus (SLE) is a chronic, multiorgan, systemic autoimmune disorder that predominantly affects women. The most common cause of death in SLE patients affected by the disease for more than five years is cardiovascular disease (CVD). A recent meta-analysis found an increased risk of stroke, myocardial infarction, CVD, and hypertension in patients, with more than a two-fold increase in relative risk (stroke: 2.51; myocardial infarction: 2.92; cardiovascular disease: 2.24; and hypertension: 2.70) [1]. Beside several nontraditional disease-specific factors such as systemic inflammation, antiphospholipid antibodies; and corticosteroid use, increased prevalence of traditional cardiovascular risk factors including hypertension, obesity, age, diabetes, and dyslipidemia also contribute to high cardiovascular event rates among patients with SLE [2,3]. Indeed, SLE is now considered to be an independent risk factor for the development of atherosclerosis [3,4].

In patients with SLE, Atta et al. discovered the prevalence of dyslipidemia to be more than 70% [5]. This dyslipidemia is characterized by elevated plasma triglyceride (TG), low-density lipoprotein cholesterol (LDL-C), and apolipoprotein B100 (ApoB100), as well as decreased plasma concentrations of high-density lipoprotein cholesterol (HDL-C) resulting in a proatherogenic lipid profile [6,7]. Previous studies reported this characteristic “lupus pattern” of lipoproteins in SLE usually occurring in the active phases of the disease [8,9,10]. Furthermore, the LDL particle size in SLE is significantly smaller than in controls [11]. Most lipid abnormalities observed in SLE might be explained by an accumulation of TG-rich lipoproteins, namely chylomicrons and very low-density lipoprotein (VLDL) particles. In their catabolism, both lipoproteins undergo degradation by lipoprotein lipase (LPL) [12]. In patients with SLE, a low LPL-activity results in the accumulation of chylomicrons and VLDLs, leading to high plasma TG and low HDL-C [13]. Beside LPL, the apolipoprotein C3–angiopoietin-like protein 4 axis is also disrupted in SLE, resulting in significantly lower ApoC3, and significantly higher ANGPLT4. Both of these are particularly important as regulators of triglyceride transport and are novel therapeutic targets [14].

To assess atherosclerosis and vasculopathy in SLE, different non-invasive, ultrasound-based imaging techniques are used. While common carotid intima-media thickness (IMT) is an early indicator of generalized atherosclerosis, brachial artery flow-mediated dilation (FMD) based on B-mode ultrasound assesses endothelium-dependent vasodilation. Furthermore, vascular stiffness is reflected by pulse-wave velocity (PWV) and augmentation index (Aix) [15]. In women with SLE, Cypiene et al. found that PWV and Aix were significantly higher, while FMD was not different from controls [16]. In adolescents with SLE, Boros et al. examined arterial stiffness and found that central PWV and characteristic impedance were elevated, while IMT, FMD, and myocardial perfusion were in the normal range [17]. However, others have detected early endothelial dysfunction indicated by low FMD in SLE [18,19,20]. In a previous study, SLE has been associated with increased arterial stiffness and higher IMT [21], while another indicated that plasma TG is an independent predictor of carotid atherosclerosis in women with SLE [22].

In summary, although several previous studies reported data on dyslipidemia and vascular diagnostic tests, their results are controversial and the link between dyslipidemia and vascular impairment in SLE has not been fully explored. Therefore, we aimed to evaluate the lipid and lipoprotein subfraction profile, inflammatory markers, and vascular diagnostic tests including arterial stiffness indicated by Aix, PWV, IMT, and FMD in active SLE to clarify the association between dyslipidemia and early vascular damage. 

We hypothesized that lipid abnormalities, especially higher concentrations of TG-rich lipoproteins, may be associated with the development of vascular dysfunction detected by the above-mentioned non-invasive vascular diagnostic tests in patients with active SLE.

## 2. Materials and Methods

### 2.1. Patient Enrollment

51 clinically active SLE patients (44 females and 7 males), who are treated at the Division of Clinical Immunology, Department of Internal Medicine, University of Debrecen, and 41 age- and gender-matched control subjects (36 females and 5 males) were enrolled in the study. Patients fulfilled the 2019 EULAR/ACR Classification Criteria for Systemic Lupus Erythematosus [23]. SLE Disease Activity Index (SLEDAI) was also calculated to stratify the activity of SLE (0 = no activity, 1–5 = mild, 6–10 = moderate, 11–19 = high, and 20 ≥ very high activity, respectively). Exclusion criteria were active lupus nephritis, pregnancy, and malignant disease. Neither the study population, nor the control group had any previous major cardiovascular event (acute myocardial infarction, ischemic stroke, or significant carotid artery stenosis). Clinical phenotypes and concomitant diseases are summarized in Appendix A. 

Patients gave informed written consent. The laboratory was approved by the National Public Health and Medical Officer Service (approval number: 094025024). The study was approved by the Regional Ethics Committee of the University of Debrecen (DE RKEB/IKEB 4775-2017, date obtained: 3 April 2020) and the Medical Research Council (ETT TUKEB 34952-1/2017/EKU, date obtained: 30 June 2017).

### 2.2. Sample Collection and Biochemical Measurements

All venous blood samples were drawn after a 12 h fasting. Routine laboratory parameters, including high-sensitivity C-reactive protein (hsCRP), total cholesterol, TG, HDL-C, LDL-C, apolipoprotein A1 (ApoA1), and ApoB100 levels were determined from fresh sera with Cobas c600 analyzers (Roche Ltd., Mannheim, Germany) from the same vendor. Anti-double-stranded deoxyribonucleic acid (dsDNA) (Orgentec, Mainz, Germany), anti-beta-2-glycoprotein I (B2GPI) (Orgentec, Germany), anticardiolipin (aCL) (Orgentec, Mainz, Germany), anti-Sm/RNP (Hycor Biomedical, Garden Grove, CA, USA), anti-Sjögren’s-syndrome-related antigen A (SSA), and anti-Sjögren’s-syndrome-related antigen B (SSB) autoantibodies (Hycor Biomedical, Garden Grove, CA, USA) were determined by enzyme-linked immunosorbent assays (ELISA). Complement 3 and 4 (C3 and C4) were measured by nephelometric methods (Siemens AG, Munich, Germany). An in-house hemolytic immunoassay was used for measuring CH50, which is a functional test for classical pathway complement activity. The commercially available methods were used following the manufacturers’ protocols. All laboratory measurements have taken place at the Department of Laboratory Medicine, University of Debrecen.

### 2.3. Interleukine-6 (IL-6) Measurement

Serum IL-6 was determined with a commercially available quantitative sandwich enzyme immunoassay technique (R&D Systems, Abington, United Kingdom). Values are expressed as pg/mL with 1.7–4.4% and 2.0–3.7% intra- and inter-assay precision, respectively. 

### 2.4. Lipoprotein Subfraction Analyses

Lipid electrophoreses were performed with a Lipoprint system for the analyses of LDL and HDL lipoprotein subfractions (Quantimetrix Corporation, Redondo Beach, CA, USA). 25 µL serum was added to the polyacrylamide gel tubes along with Sudan Black positive 300 and 200 µL loading gel solution, respectively. Tubes were photopolymerized at 21–23 °C for 30 min and then electrophoretized using a constant of 3 mA/tube in an electrophoresis chamber. Each electrophoresis chamber involved a quality control (Liposure Serum Lipoprotein Control, Quantimetrix Corporation, Redondo Beach, CA, USA). Subfraction bands were scanned with an ArtixScan M1 digital scanner (Microtek International Inc., CA, USA) and Lipoware software (Quantimetrix Corporation, Redondo Beach, CA, USA) was used for analyses. 

During LDL subfraction analysis, up to seven LDL subfractions were distributed based on their size between the VLDL and HDL peaks. Mid-C, Mid-B, and Mid-A mainly corresponded with intermediate density lipoprotein (IDL) particles (IDL-C, IDL-B, and IDL-A). Proportion of large LDL (large LDL %) was defined as the sum of the percentage of LDL1 and LDL2, whereas proportion of small LDL (small-dense LDL %) was defined as the sum of LDL3–LDL7. Cholesterol content of each LDL subfraction was calculated by multiplying the relative area under the curve (AUC) by total plasma cholesterol of the sample using Lipoware.

During HDL subfraction analysis, 10 HDL subfractions were distributed based on their size between the LDL and albumin peaks. The three major classes were calculated as the sum of HDL1–HDL3 (large HDL), HDL4–HDL7 (intermediate HDL), and HDL8–HDL10 (small HDL). Cholesterol content of each HDL subfraction was calculated by multiplying the relative AUC by total HDL-C concentration of the sample using Lipoware.

### 2.5. Flow-Mediated Vasodilation of the Brachial Artery (FMD)

Examinations were performed under standardized conditions, after 8 h of fasting and after an 18 h cessation of smoking, coffee, and tea. No vasoactive drugs were allowed 24 h prior to measurements. Tests were performed on the right brachial artery with a high-resolution duplex ultrasound using a 5–10 MHz linear transducer (Phillips HD11XE; Tampa, FL, USA). ECG was recorded during examination. We obtained a longitudinal cross-sectional image of the brachial artery 4–7 cm proximally from the cubital fossa. Arterial diameter of at least five standard points were detected. Measurements were electrocardiogram-gated and performed synchronized with R-waves [24,25,26]. After the cuff was placed on the forearm, it was inflated for 4.5 min, maintaining a 50 mmHg suprasystolic pressure above the baseline, then with a sudden release of the cuff we triggered a reactive hyperemia. Changes in the arterial diameter were detected 60 s after releasing the cuff. Mean FMD was calculated using three subsequent measurements. FMD was expressed in percentages indicating the change of diameter triggered by reactive hyperemia compared to resting diameter.

### 2.6. Determination of Carotid Intima/Media Ratio (IMT)

Measurement of IMT was also performed with duplex ultrasound using a 5–10 MHz linear transducer (Phillips, HD-11XE; Tampa, FL, USA). We examined both carotid arteries. Before taking measurements we screened the common, internal, and external carotid arteries for plaques. If no plaque formation was detectable measurements were performed 1 cm below the carotid bulb. IMT was defined by determining the distance between the first (lumen–intima border) and second (media–adventitia border) echogenic line visible in the carotid artery. We performed 10 measurements on both sides, then the average value on each side, and then the mean IMT was calculated. Results were presented in centimeters.

### 2.7. Analysis of Stiffness Parameters 

Examination was conducted under standardized circumstances after resting for 10 min. Determination of Aix and PWV was performed with an Arteriograph system (TensioMed Kft., Budapest, Hungary). The method of measurement is based on the principle that myocardial contraction produces a pulse wave in the aorta which is reflected from the aortic bifurcation. As a result, a second (reflected) wave can be observed during the systole (late systolic peak). The morphology of the second wave depends on the stiffness of the common carotid artery and on peripheral resistance which determines the amplitude of the wave. Return/reflexion time (RT S35) is calculated as the difference between the first and the second systolic wave, when cuff pressure on the brachial artery exceeds systolic pressure by at least 35 mmHg [27,28,29]. Aix can be calculated as the difference between the early and late systolic peak pressure, divided by the late systolic peak pressure. To estimate the distance between the aortic arch and bifurcation, we measured the distance between the jugular fossa and the symphysis, then used the distance to calculate PWV as the ratio of the jugulo–symphyseal distance and RT S35. Dimension of the calculated ratio is m/s. 

### 2.8. Statistical Methods

Statistical analyses were performed using the Statistica 13.5.0.17 software (TIBCO Software Inc. Palo Alto, CA, USA) and GraphPad Prism 6.01 (GraphPad Prism Software Inc., San Diego, CA, USA). The Kolmogorov–Smirnov test was used to determine the normality of data. Student’s unpaired t-test and the Mann–Whitney u-test were performed to describe the difference between continuous variables. The Chi-squared test was used to analyze the difference between binominal variables. Data were expressed as mean ± SD or median (upper quartile–lower quartile) in case of normal and non-normal distribution, respectively. Pearson’s correlation was used to analyze the relationship between continuous variables. A multiple regression analysis (backward stepwise method) was performed to determine the best independent predictor of accelerated atherosclerosis. *p* ≤ 0.05 probability values were considered statistically significant.

## 3. Results

Anthropometric and SLE-related clinical data alongside inflammatory markers and the results of imaging techniques are summarized in Table 1. Compared to controls, patients with SLE had significantly higher serum CRP and IL-6. A significantly lower Aix ratio was detected in patients compared to controls; however, there were no significant differences in IMT, FMD, and PWV between patients and controls (Table 1).

SLE patients had significantly higher plasma TG and ApoB100 concentrations, with lower HDL-C and ApoA1 compared to the control group. Higher total IDL, IDL-B, and IDL-C subfractions were found in SLE. There were no significant differences in LDL subfractions. Significantly lower concentrations of large, intermediate, and small HDL subfractions were found in patients with SLE compared to controls (Table 2).

Aix positively correlated with VLDL (r = 0.31, *p* = 0.04), IDL-C (r = 0.41, *p* = 0.006), and IDL-B subfractions (r = 0.29; *p* = 0.05) in subjects with SLE (Figure 1a–c). Marginally significant associations were found between Aix and LDL1 (r = 0.29, *p* = 0.059) (Figure 1d), TG (r = 0.27, *p* = 0.078), total cholesterol (r = 0.30, *p* = 0.058), and ApoB100 (r = 0.29, *p* = 0.057) in the patient group. Similar to Aix, PWV showed positive correlations with the concentrations of VLDL (r = 0.41, *p* = 0.007), IDL-C (r = 0.4, *p* = 0.004), IDL-B (r = 0.35, *p* = 0.02) and LDL-1 (r = 0.31, *p* = 0.04) subfractions as well as with TG (r = 0.31, *p* = 0.04) in SLE.

There were significant negative correlations between hsCRP (r = −0.4; *p* = 0.006), TG (r = −0.36; *p* = 0.02), LDL-C (r = −0.31; *p* = 0.03), ApoB100 (r = −0.34; *p* = 0.02), VLDL (r = −0.36; *p* = 0.01), LDL-2 subfraction (r = −0.32; *p* = 0.03) and FMD in patients (Figure 2).

IMT showed significant negative correlation with C4 (r = −0.4; *p* = 0.005) in patients (Figure 3a). Patients with SLE were divided into two groups based on SLEDAI; subjects with mild and moderate disease activity comprised the low (SLEDAI = 0–10), and subjects with high and very high activity comprised the high disease activity group (SLEDAI < 11). C4 was significantly higher in the low SLEDAI group (0.165 vs. 0.103 g/L; *p* = 0.03); while ApoB100 and IMT was significantly lower in the low SLEDAI group (0.82 vs. 1.03 g/L; *p* = 0.05 and 0.0478 vs. 0.0554 cm; *p* = 0.05, respectively) (Figure 3b–d).

Multiple regression analysis showed that IDL-C was an independent predictor of Aix (β = 0.399; *p* = 0.0009). The model included age, TG, total cholesterol, VLDL, IDL-C, IDL-B, and LDL1.

We could not find significant correlations between the results of vascular diagnostic test (PWV, Aix, IMT, and FMD) and lipid or inflammatory parameters in the control population.

## 4. Discussion

While LDL-C is an established major causal factor of atherosclerotic cardiovascular disease, the causality between triglyceride-rich lipoproteins, their remnants, and cardiovascular diseases is unclear [30]. Indeed, remnant particles, especially in the small VLDL and IDL range, with at least 30% cholesterol by weight, may contain up to four-fold more cholesterol molecules than an LDL particle. VLDL and remnants are also enriched in ApoE and ApoC3, both implicated in binding and retention in the artery wall [31]. These factors enhance remnant cholesterol deposition in the plaque. Moreover, unlike native LDL, which may exit the subendothelial space almost as rapidly as it enters, remnants efflux very slowly compared to their rate of entry with increased opportunity for internalization by macrophages and foam cell formation [32]. Triglyceride-rich lipoproteins affect the autophosphorylation of focal adhesion kinase and its downstream signaling pathway, phosphatidylinositol 3-kinase/protein kinase B (Akt), causing the inactivation of nitrogen-monoxide (NO) synthase (NOS), and decreased endothelial synthesis of NO. Furthermore, accumulation of triglyceride-rich lipoproteins in the plasma is associated with higher asymmetric dimethyl arginine (ADMA), an endogenous inhibitor of NOS. Moreover, these lipoproteins induce endothelin-1 (ET-1) release in humans, which induces vasoconstriction by increasing the tone of vascular smooth muscle cells, increases the proliferation of these cells, promoting thrombosis, oxidative stress, and inflammation. In addition, plasma triglyceride-rich lipoproteins increase plasma viscosity and favor a procoagulant state via increased platelet aggregation [33].

While epidemiological studies in humans suggest that remnant lipoproteins (IDL and smaller VLDL) are predictors of the severity or progression of atherosclerosis [34], their causal role in the enhanced atherogenesis of patients with SLE has not been clarified. A previous study reported that atherogenic lipoproteins, including small VLDL subsets, were associated with increased disease activity, and the ApoB100:ApoA1 ratio correlated positively with SLEDAI disease activity score in juvenile SLE [35]. We found higher serum ApoB100 in patients with high/very high SLEDAI compared to those with mild/moderate disease activity indicating the role of inflammatory processes in disturbed lipoprotein metabolism. Moreover, we also detected higher VLDL and smaller IDL subfractions in our young patients with clinically active SLE.

Although Aix is probably not a commonly mentioned arterial stiffness parameter, in our study Aix was the only vascular diagnostic parameter that showed significant alteration in our young patients with SLE. Its significant positive correlations with VLDL, IDL-B, and IDL-C may also support the atherogenic potential of these triglyceride-rich lipoproteins in the early phase of atherosclerosis. Multiple regression analysis also showed that IDL-C was an independent predictor of Aix, which also highlights the probable importance of triglyceride-rich lipoproteins in the pathomechanism of SLE-associated vascular complications. In a former study, Aix turned out to be related to organ damage measured by Systemic Lupus International Collaborative Clinics (SLICC) index in young women diagnosed with SLE without significant organ damage [16]. In another study, higher Aix was reported in a middle-aged SLE cohort, and Aix correlated with IMT in the common carotid artery, common femoral artery, and internal carotid artery [36]. A recent meta-analysis comprising 49 studies also showed increased Aix in patients with SLE compared to healthy controls [37], which may also underline the importance of measuring Aix.

While IMT and stiffness are relatively stable, FMD may be influenced by many confounding factors [15]. We found that IMT was higher in patients with high/very high disease activity compared to those with mild/moderate, which demonstrates the crucial role of inflammation in atherosclerotic remodeling of the arterial wall even in younger ages. Although we did not find significant differences in FMD values between patient with SLE and controls or between patients with high or low disease activity, significant negative correlations were found between serum hsCRP, TG, LDL-C, ApoB100, VLDL, LDL2, and FMD. These results are in line with the above-mentioned multicausal characteristics of FMD.

A previous study demonstrated that increased carotid IMT in both the right and left carotid arteries and increased PWV in the left carotid artery of patients with SLE correlated positively with the increase in LDL subfraction L5 percentage (representing electronegative, small-dense LDL) determined by a fast protein liquid chromatography system [38]. Despite the larger patient population, we could not find any significant correlations between PWV and LDL subfractions, probably due to the different method used for LDL subfraction analysis and the lower age of our patients.

Like other previous studies, we could not find any significant correlations between vascular index and lipid markers in the control population, probably because of the lack of vascular damage in these young, mostly female subjects [20,38]. The lack of associations between the results of vascular diagnostic tests (PWV, Aix, IMT, and FMD) and lipid or inflammatory parameters in the control population may also indicate that vascular abnormalities found in patients with SLE are disease-specific and due to the SLE-associated dyslipidemia and inflammation.

Beside LDL-C reduction, 3-hydroxy-3-methyl-glutaryl-CoA (HMG-CoA) reductase inhibitor statins exert pleiotropic effects to modulate various components of the immune system; hence, they might be of benefit in SLE by inhibiting immune activation in the arterial wall and attenuating disease activity [39]. However, some former uncontrolled studies reported a few cases of lupus-like syndrome, autoimmune hepatitis, skin lesions similar to subacute lupus, and dermatomyositis related to statin administration [40]. Furthermore, the effect of statins on triglyceride-rich lipoproteins is limited. Taken together, statin administration might not be the optimal lipid-lowering strategy in SLE. On the other hand, well-known and novel agents lowering triglyceride-rich lipoproteins are currently being tested for the prevention of atherosclerotic cardiovascular disease in large clinical trials, including REDUCE-IT, STRENGTH, and PROMINENT using purified omega-3 fatty acid and selective peroxisome proliferator-activated receptor α modulator pemafibrate, with inconclusive results. However, even more studies appeared on the horizon aiming to inhibit apolipoprotein C3 and angiopoietin-related protein 3 and 5 [41]. Promising results demonstrating substantial lowering of triglyceride-rich lipoproteins have already been reported after administrating some of these drugs in development; thus, it is possible that therapies aiming at lowering triglyceride-rich lipoproteins could become increasingly available in the clinical setting [41], and, based on our results, SLE-associated dyslipidemia should be considered as a potential new indication.

Some limitations of our study should also be mentioned. The relatively small number of SLE patients may reduce the power of our study. The low number of male patients does not allow the investigation of gender differences. It must be noted that cardiovascular diseases occur independently of SLE. Since our research groups (both SLE patients and healthy controls) were young, and mostly females, the risk of any cardiovascular events is very low. Therefore, vascular abnormalities identified by vascular diagnostic tests are likely the consequences of SLE-associated dyslipidemia and systemic inflammation. Furthermore, long-term follow-up could help us clarify the role of triglyceride-rich lipoproteins in the atherosclerotic processes associated with SLE. Still, these results underline the potential importance of studying the detailed lipoprotein profile including triglyceride-rich lipoproteins and the measurement of arterial stiffness parameters in this special patient population.

## 5. Conclusions

This is the first clinical study evaluating the lipid and lipoprotein subfraction profile, inflammatory markers, and various vascular diagnostic tests including arterial stiffness, IMT, and FMD simultaneously in young, clinically active SLE patients. Our results demonstrate the possible effect of triglyceride-rich lipoproteins on vascular function in young SLE patients and underline the importance of further studies to clarify its long-term consequences. Based on our results measurement of Aix may be used for the early detection of SLE-associated vascular abnormalities. Given the high risk of cardiovascular morbidity and mortality in young patients with SLE, it is advisable to routinely assess risk factors including triglyceride-rich lipoproteins and arterial stiffness parameters and to integrate appropriate preventive measures into our patients’ complex management plans.

## Figures and Tables

**Figure 1 biomolecules-13-00401-f001:**
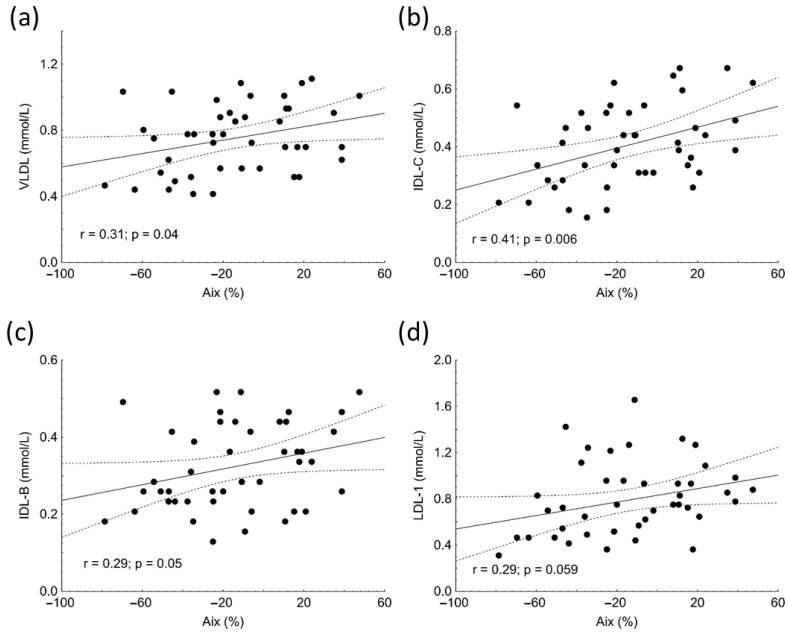
Correlations between levels of very low-density lipoprotein (VLDL) (**a**), intermediate density lipoprotein (IDL-C) (**b**), IDL-B (**c**), low-density lipoprotein-1 (LDL1) (**d**) subfractions and augmentation index (Aix) in patients with systemic lupus erythematosus (SLE).

**Figure 2 biomolecules-13-00401-f002:**
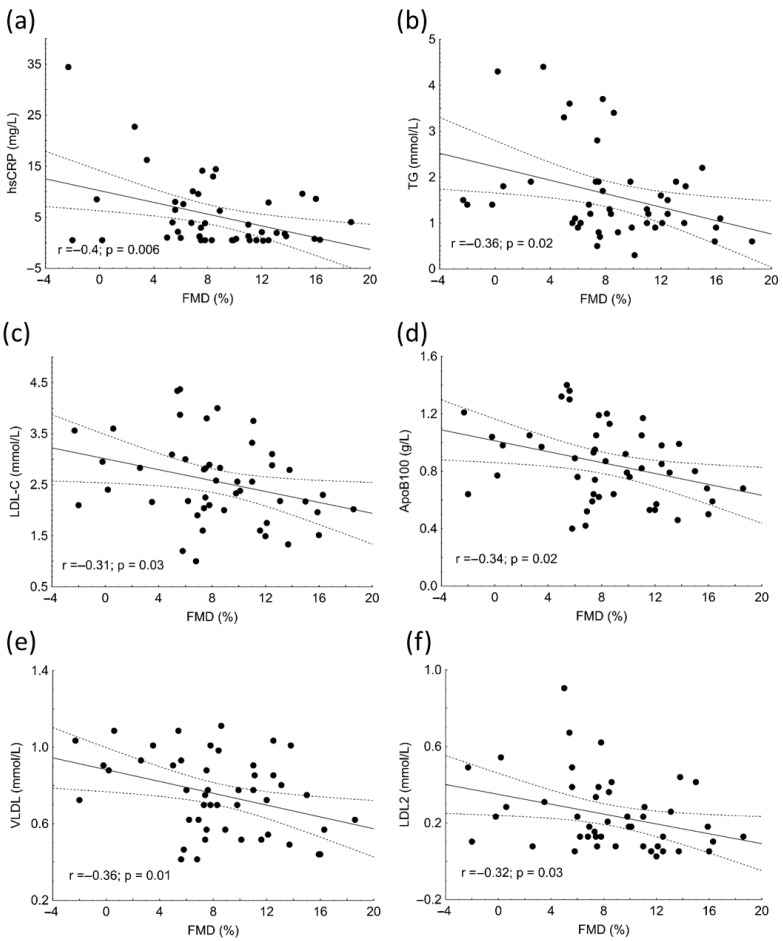
Correlations between high-sensitivity C-reactive protein (hsCRP) (**a**), triglyceride (TG) (**b**), low-density lipoprotein cholesterol (LDL-C) (**c**), apolipoprotein B (ApoB) (**d**), very low-density lipoprotein (VLDL) subfraction (**e**), LDL2 subfraction (**f**) and flow-mediated dilation (FMD) in patients with systemic lupus erythematosus (SLE).

**Figure 3 biomolecules-13-00401-f003:**
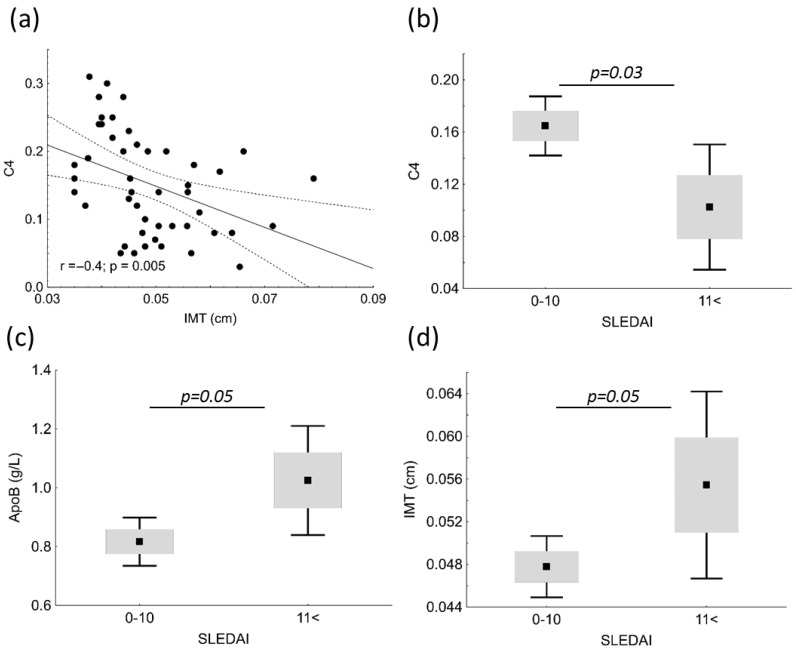
Correlations between complement factor 4 (C4) and carotid intima media thickness (IMT) (**a**) in patients with systemic lupus erythematosus (SLE). C4 (**b**), apolipoprotein B (ApoB) (**c**) and IMT (**d**) in patients with mild/moderate SLE disease activity (SLEDAI = 0–10) and with high/very high SLEDAI (SLEDAI < 11).

**Table 1 biomolecules-13-00401-t001:** Anthropometric and clinical data, inflammatory parameters, and vascular damage tests of study individuals. Values are presented as mean ± standard deviation or median (lower quartile–upper quartile).

	Patients with SLE	Controls	*p*-Value
Number of subjects	51	41	
Female/male	44/7	36/5	ns.
Age (years)	31.82 ± 6.4	31.4 ± 7.2	ns.
BMI (kg/m^2^)	23.7 ± 4.3	23.8 ± 7.4	ns.
Smoking (%)	17 (33.3)	13 (31.7)	ns.
Hypertension (%)	2 (3.9)	3 (7.3)	ns.
Diabetes mellitus (%)	1 (1.96)	0	ns.
SLEDAI	5.96 (2–10)		na.
APS (%)	3 (5.9)		na.
aPL antibodies (%)	7 (13.7)		na.
prednisolone (%)	51 (100)		na.
chloroquine (%)	25 (49)		na.
NSAIDs (%)	17 (33.3)		na.
DMARDs (%)	32 (62.7)		na.
hsCRP (mg/L)	2.13(0.59–7.88)	1.07(0.05–2.23)	*p* < 0.05
IL-6 (mg/L)	0.54 (0–3.25)	0 (0–0.17)	*p* < 0.01
C3 (g/L)	0.92 ± 0.28		na.
C4 (g/L)	0.15 ± 0.08		na.
CH50 (g/L)	42.5 ± 16.4		na.
IMT (cm)	0.049 ± 0.01	0.037 ± 0.004	ns.
FMD (%)	8.43 ± 4.8	9.32 ± 3.2	ns.
Aix (%)	−13.95 ± 31.6	−48.66 ± 22	*p* < 0.05
PWV (m/s)	7.88 ± 1.5	8.2 ± 18	ns.

Abbreviations: Aix: augmentation index; aPL: antiphospholipid antibodies; APS: anti-phospholipid syndrome; BMI: body mass index; C3: complement 3; C4: complement 4; CH50: 50% hemolytic complement; IMT: carotid intima-media thickness; hsCRP: high sensitivity C-reactive protein; DMARDs: disease-modifying anti-rheumatic drugs; FMD: flow-mediated dilation; IL-6: interleukine-6; na.: not applicable; ns.: not significant; NSAIDs: non-steroid anti-inflammatory drugs; PWV: pulse wave velocity; SLE DAI: Systemic Lupus Erythematosus Disease Activity Index.

**Table 2 biomolecules-13-00401-t002:** Concentrations of lipid parameters and lipoprotein subfractions in study participants. Values are presented as mean ± standard deviation or median (lower quartile–upper quartile).

	Patients with SLE	Controls	*p*-Value
Total cholesterol (mmol/L)	4.57 ± 1.1	4.74 ± 0.8	ns.
LDL-C (mmol/L)	2.59 ± 0.8	2.67 ± 0.7	ns.
HDL-C (mmol/L)	1.24 ± 0.4	1.71 ± 0.4	<0.001
Triglyceride (mmol/L)	1.25 (0.9–1.9)	0.8 (0.7–1.4)	<0.01
ApoA1 (g/L)	1.39 ± 0.4	1.63 ± 0.4	<0.01
ApoB100 (g/L)	0.86 ± 0.3	0.73 ± 0.2	<0.01
Lipoprotein(a) (mg/L)	106 (80–294)	91 (80–181)	ns.
**Lipoprotein subfractions**			
VLDL (mmol/L)	0.75 ± 0.2	0.71 ± 0.1	ns.
Midband (IDL) (mmol/L)	0.59 ± 0.2	0.84 ± 0.2	<0.001
IDL-A (mmol/L)	0.59 ± 0.2	0.58 ± 0.2	ns.
IDL-B (mmol/L)	0.33 ± 0.1	0.28 ± 0.1	*p* < 0.05
IDL-C (mmol/L)	0.41 ± 0.1	0.35 ± 0.1	*p* < 0.05
** *LDL subfractions* **			
LDL-1 (mmol/L)	0.79 ± 0.3	0.90 ± 0.2	ns.
LDL-2 (mmol/L)	0.25 ± 0.2	0.24 ± 0.1	ns.
LDL-3 (mmol/L)	0.03 ± 0.1	0.02 ± 0.01	ns.
LDL-4 (mmol/L)	0.002 ± 0.01	0	ns.
LDL-5 (mmol/L)	0	0	ns.
** *HDL subfractions* **			
Large HDL (mmol/L)	0.48 ± 0.3	0.64 ± 0.3	<0.01
Intermediate HDL (mmol/L)	0.59 ± 0.2	0.84 ± 0.2	<0.001
Small HDL (mmol/L)	0.16 ± 0.1	0.25 ± 0.1	<0.001

Abbreviations: ApoA1: apolipoprotein A1; ApoB100: apolipoprotein B100; HDL: high-density lipoprotein; IDL: intermediate-density lipoprotein; LDL: low-density lipoprotein; VLDL: very low-density lipoprotein.

## Data Availability

All data generated or analyzed during this study are included in this published article. All data generated or analyzed during the current study are available from the corresponding author on reasonable request.

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
