# Peer review of "Role of Altered Metabolism of Triglyceride-Rich Lipoprotein Particles in the Development of Vascular Dysfunction in Systemic Lupus Erythematosus"

_biomolecules, 2023, doi:10.3390/biom13030401_

Round 1

Reviewer 1 Report

In the presented manuscript the authors evaluate the lipid profile, inflammatory markers, and vascular diagnostic tests in active SLE patients to clarify the association between dyslipidemia and early vascular damage. The authors proved that in young patients with SLE triglyceride-rich lipoproteins influence vascular function detected by Aix. These parameters may be assessed and integrated into the management plan for screening cardiovascular risk in patients with SLE. In my opinion, the level of performed research is significant and valuable. The manuscript is well-planned and written. The experimental part is well prepared with attention to detail.

The main comments

1. Due to the specificity of the conducted research, many abbreviations were used in the manuscript but all abbreviations were explained the first time they were used. It is worth making a list of used abbreviations at the end of the work to facilitate the analysis of the presented data.

2. Do the authors have any information about other diseases of patients, medications taken, and addictions that could affect the results of the presented studies? Could it have turned out that cardiovascular diseases occur independently, not necessarily in connection with SLE?

Author Response

Dear Reviewer, thank you for your positive reply and helpful comments on our manuscript.

We hereby answer your recommendations as follows:

  1. Due to the specificity of the conducted research, many abbreviations were used in the manuscript but all abbreviations were explained the first time they were used. It is worth making a list of used abbreviations at the end of the work to facilitate the analysis of the presented data.

Response: Thank you for the comment. We checked all abbreviations and explained the missing ones the first time they were used (Ln 60-60, 125, 227, 300-301). Furthermore, we made a list of abbreviations and added it to the end of the manuscript (Ln 426-464).

  1. Do the authors have any information about other diseases of patients, medications taken, and addictions that could affect the results of the presented studies? Could it have turned out that cardiovascular diseases occur independently, not necessarily in connection with SLE?

Response: Thank you for this forward-looking comment. Yes, we have some further data on clinical phenotypes and other concomitant diseases. We demonstrated the most important ones in Supplementary Table 1. Furthermore, we added NSAID use to table 1. However, because of the low number of cases with other diseases we believe that these factors might not influence our results significantly.

In general, the reviewer is right, these factors may also alter the risk of cardiovascular diseases, and we cannot exclude the possibility of the occurrence of cardiovascular events that may be independent of SLE. It must be noted that the mean age of our patients and controls were 31.82±6.4 and 31.4±7.2 years. Fortunately, the cardiovascular risk of young female subjects, especially under 32 years of age, in the general population is very low (see figure below, which demonstrates mortality rates for coronary heart disease in the USA in 2000 by sex; Global Burden of Disease study data). Therefore, we believe that we don’t have to account for this bias. Still, we completed the limitations with this issue (Ln 378-382).

The following sentences were added:

“It must be noted that cardiovascular diseases occur independently of SLE. Since our research groups (both SLE patients and healthy controls) were young, and mostly female, the risk of any cardiovascular event is very low. Therefore, vascular abnormalities identified by vascular diagnostic tests are likely the consequences of SLE-associated dyslipidemia and systemic inflammation. “

Again, we are very thankful for your valuable and thorough review.

Reviewer 2 Report

In this article, the authors investigate the relationship between labo data and vascular tests to clarify the cause of vascular damage in SLE.

Regarding the correlation analysis, the correlations were very low in all analyses, where the relationship might be low. The authors may show the regression analysis data to understand the total findings. In that case, are many factors correlated with each other? 

The authors analyzed the relationship between the vascular index and lipid marker in SLE, but I am interested in the results of it in controls. Is the relationship seen specifically in SLE? Are there some methods to show the specificity compared to normal or other disease groups?

Author Response

Response to Reviewer 2:

Dear Reviewer, thank you for your helpful comments and suggestions on our manuscript. We hereby answer your recommendations as follows:

Regarding the correlation analysis, the correlations were very low in all analyses, where the relationship might be low. The authors may show the regression analysis data to understand the total findings. In that case, are many factors correlated with each other?

Response: Thank you for the valuable comment. Indeed, we made many correlation analyses, but we included only the significant results to highlight them. The reviewer is right; therefore, we included the results of all the other analyses, and summarized their results in Supplementary Table 2.

Multiple regression analysis for Aix was performed and included (Ln 273-275), which showed that IDL-C was an independent predictor of Aix (β=0.399; p=0.0009). The model included age, TG, total cholesterol, VLDL, IDL-C, IDL-B and LDL1 since these parameters correlated significantly with Aix. All other correlations were not significant. Therefore, those parameters are not predictors, and were not included in the backward stepwise multiple regression analysis.

Detailed results of the backward stepwise multiple regression analysis are uploaded in a separate file. 

The authors analyzed the relationship between the vascular index and lipid marker in SLE, but I am interested in the results of it in controls. Is the relationship seen specifically in SLE? Are there some methods to show the specificity compared to normal or other disease groups?

Response: Thank you for the comment. Like other previous studies, we could not find any significant correlation between vascular index and lipid markers in the control population, probably because of the lack of vascular damage in these young (mean age was <32 ys), mostly female subjects. Based on these findings, we believe that the relationship is SLE-specific, due to the SLE-associated dyslipidemia and inflammation. However, further multicenter studies on large patient populations would be needed to calculate the specificity of the imaging methods compared to healthy or other disease groups.

Data on control subjects and their explanation were added to the Results and Discussion sections.

The following sentences were added:

“We could not find significant correlations between the results of vascular diagnostic tests (PWV, Aix, IMT and FMD) and lipid or inflammatory parameters in the control population.” (Ln 276-278)

“Like other previous studies, we could not find any significant correlations between vascular index and lipid markers in the control population, probably because of the lack of vascular damage in these young, mostly female subjects. The lack of association between the results of vascular diagnostic tests (PWV, Aix, IMT and FMD) and lipid or inflammatory parameters in the control population may indicate that the vascular abnormalities found in patients with SLE are disease-specific and due to the SLE-associated dyslipidemia and inflammation.” (Ln 350-356)

Again, we are very thankful for your valuable and thorough review.

Round 2

Reviewer 2 Report

The authors tried to answer my questions individually, and the manuscript improved well.